# Personality in motion: How intuition and sensing personality traits relate to lower limb rebound performance

Cyrille Gindre[1,2,3,4☯], Aurélien Patoz [2,5☯] *, Bastiaan Breine[2,6☯], Thibault Lussiana[1,2,3,4☯]

**1** Research and Development Department, Volodalen, Chavéria, France, **2** Research and Development Department, Volodalen SwissSportLab, Aigle, Switzerland, **3** MPFRPV, Université de Franche-Comté, Besançon, France, **4** Exercise Performance Health Innovation (EPHI) Platform, Besançon, France, **5** Institute of Sport Sciences, University of Lausanne, Lausanne, Switzerland, **6** Department of Movement and Sports Sciences, Ghent University, Ghent, Belgium

☯ These authors contributed equally to this work.
\* aurelien.patoz@unil.ch

**Data Availability Statement:** The datasets for this study are freely available using the access link https://github.com/aurelienPatoz/we-run-the-way-we-are.

## Abstract

Embodied cognition asserts a symbiotic relationship between cognitive processes and the physical body, raising an intriguing question: could personality traits be intertwined with the biomechanical performance of the lower limb? This study aimed to explore this connection by examining how personality traits, assessed using the Myers-Briggs Type Indicator (MBTI), relate to lower limb rebound power (RP) measured through the five-repetition rebound jump test. Eighty participants completed two sessions: a biomechanical analysis of hopping using an Optojump® system to measure contact time, flight time, and RP, and a personality traits assessment categorizing traits across four MBTI axes: extraversion-introversion (favorite world); sensing-intuition (information processing preference); thinking-feeling (decision making); and judging-perceiving (structure). Participant characteristics did not significantly differ across MBTI axes ($p \geq 0.07$), minimizing potential confounding factors. Notably, individuals classified as intuitive showed significantly longer flight times ($p = 0.02$) and larger RP ($p = 0.007$) compared to sensing individuals, suggesting a greater reliance on the fast stretch-shortening cycle and showcasing superior use of their lower limb structures as springs. This suggests potential implications for sports performance, with intuition individuals possibly excelling in plyometric sports. However, no significant associations were found between biomechanical performance and the other three MBTI axes ($p \geq 0.12$), challenging the initial hypothesis. This research provides initial insights into the nuanced relationship between personality traits and movement patterns, indicating the potential for tailored physical interventions to enhance adherence and optimize responses in training programs.

## Introduction

Traditional perspectives that separate the mind from the body are challenged by embodied cognition [1], which proposes to connect the physical body and cognitive processes. This

**Funding:** The author(s) received no specific funding for this work.

**Competing interests:** The authors have declared that no competing interests exist.

accentuates the profound impact of motor and sensory experiences on mental functions [2]. Embodied cognition acknowledges the pivotal role of the body's engagements with the environment in shaping cognitive phenomena, departing from conventional beliefs that confine cognition to the brain [3]. This framework underscores both top-down processes, where higher-order cognitive functions mold the perception and interaction with the surrounding world and bottom-up processes, where sensory input informs cognitive operations.

The reciprocal interplay among the body, mind, and the dynamic world they collectively traverse is unveiled by delving into the intricate dynamics of embodied cognition. Indeed, research showed that both the level of physical activity [4] and the walking speeds [5, 6] increased when conscientiousness and extraversion are set to higher levels. Another study extended the evidence of an association between personality and physical function to include its strength component, indicating health behavior as an important pathway [7]. Additionally, high intellectual curiosity was linked to establishing exercise habits in high school, suggesting that tailored exercise promotion can help prevent declines in muscle mass and motor function [8]. Furthermore, research showed that during walking, the level of extraversion links with the quantity of thorax motion for male while the level of agreeableness links with the level of pelvis motion for female [9]. Moreover, there are indications that personality traits may influence a runner's likelihood of engaging in barefoot running [10]. Recently, gait analysis recorded using smartphone sensors [11] or videos [12] was used to classify personality traits using machine learning. Finally, personality traits were associated with distinct running biomechanics [13]. Overall, personality traits may mirror the individuals' movement patterns. These personality traits could be effectively classified into one of two possible categories along various axes using the Myers-Briggs Type Indicator (MBTI) test, a tool rooted in Jung's psychology [14]. The MBTI was also utilized in the field of sport psychology assessment, primarily for team cohesion and increasing awareness of the relationships between teammates and coaches [15].

The intricate interplay between the mind and body prompts an intriguing question. Could personality traits be intertwined with the biomechanical performance of the lower limb?

Vertical jumps, widely used by strength and conditioning professionals, coaches, and healthcare practitioners, serve as performance tests for evaluating lower limb muscular strength [16], highlighting the significance of this question. The plyometric capacity of the lower limb can be evaluated through the five-repetition rebound jump test [17, 18], rooted in the fast stretch-shortening cycle characterized by an eccentric contraction followed by an immediate (short coupling time; $\leq 250$ ms) concentric contraction of the muscle [19]. Individuals who execute repetitive jumps as high and as quickly as possible, displaying the greatest rebound power (RP, W/kg), are anticipated to excel in plyometric sports [20].

Hence, the present study explored whether the RP of the lower limb, assessed by the five-repetition rebound jump test, could be related to personality traits. The hypothesis posited that the RP of the lower limb would be associated with personality traits.

## Materials and methods

### Experimental approach to the problem

Each participant completed two experimental sessions held within one week: a hopping biomechanical analysis during the first session, and a personality traits assessment during the second session. The biomechanical analysis permitted to assess RP in a non-specific running task to quantify the ability of individuals to generate force in a plyometric mode. Personality traits were obtained using the MBTI test.

## Participants

Eighty individuals, 13 females (age: 30 ± 12 years, height: 167 ± 7 cm, body mass: 61 ± 9 kg, and training hour per week: 9 ± 8 h/week) and 67 males (age: 29 ± 11 years, height: 178 ± 6 cm, body mass: 72 ± 9 kg, and training hours per week: 6 ± 4 h/week), voluntarily participated in this study. The university's institutional review board (Comité de Protection des Personnes Est 1 (CPP EST 1)) approved the protocol prior to participant recruitment (ID RCB 2014-A00336-41). For study inclusion, participants were required to be in good self-reported general health with no current or recent (<6 months) musculoskeletal injuries. Participants were recruited between the September 1st and November 30th of 2014. Each participant provided written informed consent after a comprehensive explanation of the study's nature and potential consequences.

## Procedures

**Assessment of rebound power.**  A 10-min warm-up run was performed at a self-selected speed (range: 2.5–3.5m/s) on an indoor athletic track. Then, RP, i.e., the plyometric characteristics of the lower limbs [21, 22], was assessed by asking participants to perform the five-repetition rebound jump test. All jumps were performed with hands placed on the hips and participants were wearing their habitual training shoes. Participants were required to jump vertically as high and as fast as possible while minimizing ground contact time ($t_c$) and maximizing flight time ($t_f$). Participants were also instructed to minimize knee actions (i.e., flexion and extension) during the test. The following instructions were provided before the test: "Upon the acoustic signal, perform an initial countermovement jump (impetus), and complete six hops in place, with minimal knee flexion and a maximal jump height. After the sixth jump, remain in a vertical standing posture and wait for the final acoustic signal.". Multiple supervised trials were performed to familiarize participants with this hopping task and to make sure the task was properly executed. The hops were recorded once the researcher deemed the technique satisfactory using an Optojump® system (MicroGate Timing and Sport, Bolzano, Italy). The Optojump® system, connected to a personal computer, consists of two parallel bars placed 1 m apart. One bar contains light-emitting diodes positioned 3 mm above the ground, acting as the transmitter unit, while the other bar serves as the receiver unit. Every millisecond (sampling frequency of 1000 Hz), the system checks the status of the light transmission. If the light is blocked by an individual's feet during jumping, a designated variable is set to 1. Conversely, if the light reaches the receiver, the variable is set to 0. Therefore, $t_c$ (ms) was given by the time during which the light was blocked (variable is 1), and $t_f$ (ms) by the time during which the light was transmitted (variable is 0). Then, RP for each hop was calculated following the method described in Bosco, Luhtanen (22) using the following equation (Eq 1).

$$\text{RP} = \frac{g^2 t_f (t_f + t_c)}{4 t_c} \tag{1}$$

Given that the first hop served as an impetus, the average of the five remaining hops was used to calculate RP. Three repetitions of the test were performed with a 30 s rest. The best repetition, based on the highest RP was used for statistical analysis, to ensure the reliability and validity of the present findings in assessing RP. This approach allowed selecting the most representative performance data, minimizing the influence of variability that may occur in individual trials and provided a more accurate reflection of participants' lower limb biomechanics during vertical jumping. The Optojump® system has an excellent test-retest reliability with high intraclass correlation coefficients (range: 0.982–0.989) and low coefficients of variation (2.7%) [23].

**Assessment of personality traits.** Personality traits were classified in one of two possible categories along the four MBTI axes: extraversion-introversion (favorite world); sensing-intuition (information processing preference); thinking-feeling (decision making); and judging-perceiving (structure), based on the answers to 93 questions. The way a person perceives a situation and decides on a course of action is influenced by each of these four axes. Over 4 weeks, an agreement of 84 to 96% with a median of 90% was reported for each dichotomy [14]. Furthermore, each of the MBTI axes is showing an excellent stability. Indeed, the test-retest correlations were between 0.83 and 0.97 over a 4-week interval, which is higher than that of many established trait measures, and between 0.77 and 0.84 over a 9-month interval [24]. To ensure the quality of the questionnaire data, the personality traits of participants were reassessed by a MBTI-certified practitioner via a face-to-face meeting (~1 h), especially because the results of the MBTI seem to be context-dependent [24].

## Statistics

Assuming a power of 0.8, an α error of 0.05, and a moderate effect sizes (~0.5) for the difference in RP between MBTI axes, 80 participants were required as determined by sample size calculation [25]. G\*Power (v3.1, available at https://www.psychologie.hhu.de/arbeitsgruppen/allgemeine-psychologie-und-arbeitspsychologie/gpower) was used to obtain the sample size [26]. Descriptive statistics are presented as mean ± standard deviation. Levene and Kolmogorov-Smirnov tests were used to evaluate homogeneity of variance and data normality, respectively. ANOVA and non-parametric ANOVA when data normality was not verified were used to compare participant characteristics along each MBTI axis. ANOVA and pair-wise post-hoc comparisons employing Holm corrections were used to investigate the effect of each MBTI axis on participant characteristics, $t_c$, $t_f$, and RP. For each significant comparison, Cohen's $d$ effect size was calculated and classified as *small*, *moderate*, and *large* when $|d|$ values were larger than 0.2, 0.5, and 0.8, respectively. Jamovi (v1.6.23, https://www.jamovi.org), with a level of significance set at α ≤ 0.05, was used to perform statistical analysis.

## Results

Table 1 reports the classifications of participants along each MBTI axis. Homogeneity of variance was verified for all participant characteristics (age, height, body mass, and weekly training hours; $p \geq 0.31$). Normality was verified for height and body mass ($p \geq 0.23$) while age and weekly training hours were not normally distributed ($p \leq 0.04$). No main effect of the MBTI axes on age, height, body mass, and weekly running hours ($p \geq 0.07$) were reported by ANOVA and non-parametric ANOVA. Hence, each MBTI axis reported similar participant characteristics.

**Table 1. Participants within the four Myers-Briggs Type Indicator (MBTI) axes.**

| MBTI axis | Group | Number of participants |
|---|---|---|
| Extraversion-introversion | Extraversion | 37 (46%) |
| | Introversion | 43 (54%) |
| Sensing-intuition | Sensing | 47 (59%) |
| | Intuition | 33 (41%) |
| Thinking-feeling | Thinking | 35 (44%) |
| | Feeling | 45 (56%) |
| Judging-perceiving | Judging | 41 (51%) |
| | Perceiving | 39 (49%) |

**Table 2. Contact time ($t_c$), flight time ($t_f$), and rebound power (RP) of the lower limb measured while hopping in place for sensing and intuition individuals.**

| Biomechanical variable | Sensing-intuition group | Value |
|---|---|---|
| $t_c$ (ms) | Sensing | 189 ± 19 |
| | Intuition | 181 ± 21 |
| $t_f$ (ms) | Sensing | 415 ± 51* |
| | Intuition | 451 ± 53 |
| RP (W/kg) | Sensing | 34 ± 7* |
| | Intuition | 40 ± 8 |

Note: values are presented as mean ± standard deviation.

* Significant difference ($p \leq 0.05$) between sensing and intuition individuals as determined by Holm post-hoc tests.

Homogeneity of variance and normality were verified for $t_c$, $t_f$, and RP ($p \geq 0.34$). The ANOVA reported an effect of the MBTI axes for $t_f$ and RP ($p \leq 0.01$) but not for $t_c$ ($p = 0.54$). Pair-wise post-hoc comparisons indicated that intuition individuals had a longer $t_f$ ($p = 0.02$; Table 2) and larger RP ($p = 0.007$; Fig 1 and Table 2) than sensing individuals with *moderate* effect sizes ($|d| \geq 0.69$). Otherwise, there was no difference in $t_f$ and RP among the other three MBTI axes ($p \geq 0.12$).

## Discussion

The focus was to understand how cognition, as categorized by the MBTI test, may influence the way individuals jump. Differences in the RP of the lower limb were reported between sensing and intuition individuals, supporting our hypothesis. Intuition individuals demonstrated a larger RP than sensing individuals, suggesting that these individuals have a greater reliance on the fast stretch shortening cycle and are thus better able to use their lower limb structures as springs. However, there was no association between the other three MBTI axes and the RP of the lower limb, partly refuting our hypothesis.

The objective power measure extracted from the five-repetition rebound jump test, i.e., hopping in place as high and as fast as possible, supported the greater reliance of intuition individuals on the fast stretch shortening cycle than sensing individuals (Fig 1 and Table 2). The significant difference observed in RP suggest that the re-use of elastic energy is an inherent feature of intuition individuals and that these individuals are better able to use their lower limb structures as springs, which is one of the multiple functional roles of the musculoskeletal system [27]. The larger RP of intuition compared to sensing individuals was associated with a longer $t_f$ but no difference in $t_c$ (Table 2). Worth noting is that intuition and sensing individuals shared similar participant characteristics ($p \geq 0.07$), thus removing potential confounding variables [28, 29]. However, many other personal characteristics not measured herein could also confound the present results. For instance, the elastic strategy was shown to be better supported by smaller moment arms of the Achilles tendon than longer moment arms [30]. Therefore, lower limb anatomy and its relation to the RP of the lower limb could also be explored by future experimentations.

A better quality of the repetitive vertical jumps, i.e., a RP as large as possible, was related to a better plyometric capacity [17]. Herein, those individuals were associated with an intuition personality trait (Fig 1 and Table 2). Intuition individuals tend to focus on the meaning and patterns of information. They gravitate towards abstract concepts and theories, effortlessly making unconscious connections across various domains of knowledge [14]. In contrast,

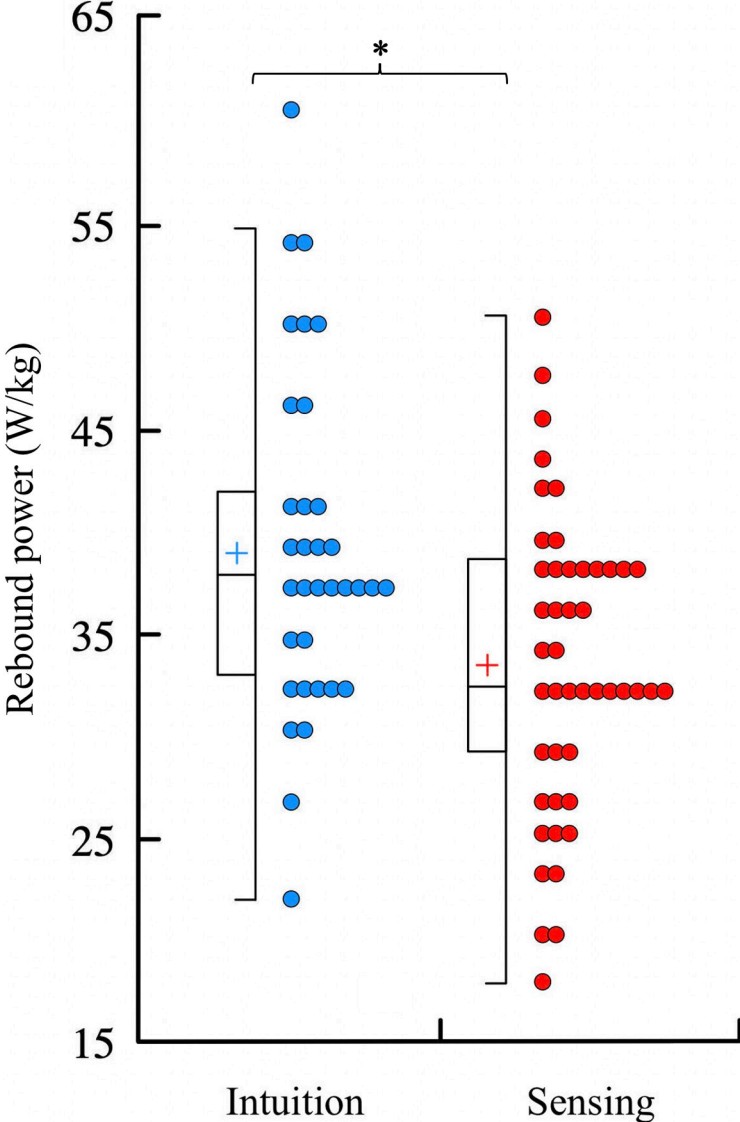

**Fig 1. Rebound power of the lower limb measured while hopping in place for intuition and sensing individuals.** *
Significantly larger ($p \leq 0.05$) rebound power for intuition (blue boxes; left side) than sensing individuals (red boxes;
right side) as determined by Holm post-hoc tests.

sensing individuals direct their attention to physical realities, showing a preference for practi-
cal and specific facts that can be directly perceived through their physical senses [14]. This
investigation unveiled that a penchant for abstract concepts correlates with a larger RP.
According to Eq (1), a larger RP can be obtained by decreasing $t_c$ and/or increasing $t_f$, indicat-
ing a more dynamic rebound pattern. Hence, intuition individuals might perform better than
sensing individuals in plyometric sports while sensing individuals would be more suitable for
sports that do not require this characteristic. Therefore, the choice of sport could be due to
both the intrinsic qualities and the intuition-sensing personality trait of individuals. In other
words, it would be more difficult to improve performance in a chosen sport if the intuition-
sensing personality trait does not match the characteristics of this sport. A similar idea was

previously suggested based on the spontaneous running pattern adopted by young soccer players [31]. Indeed, the authors suggested that soccer players classified as terrestrial runners were more adapted to the specific short distance and high acceleration sprints than those classified as aerial runners. Similarly, plantar flexors' maximal rate of force development was significantly higher in aerial than in terrestrial runners, while maximal voluntary contraction force remained unchanged [32]. This was accompanied by higher myoelectrical activity recorded in the soleus muscle. While M-waves and other parameters remained unchanged, maximal H-reflex was significantly higher in aerial than in terrestrial runners, still in soleus only. The findings of this previous study raised the possibility of different plantar flexors' neuromuscular characteristics according to running profile which were focused on the soleus rather than on the gastrocnemii and demonstrated a close link between running forms and neuromuscular and mechanical variables [32].

The study focused on specific measures such as $t_c$, $t_f$, and RP due to their well-established relevance and reliability in assessing lower limb biomechanical performance [33–35]. By concentrating on these particular metrics, we aimed to ensure that our findings would be grounded in robust and widely accepted performance indicators. While it is acknowledged that the relationship between personality traits and biomechanical performance can and should be more complex, the selection of these specific measures allowed us to maintain a clear and focused scope for the study. This approach provided a solid foundation for exploring initial relations between the MBTI axes and biomechanical performance of the lower limb. Moreover, the simplicity of these measures does not preclude the existence of more nuanced interactions. Instead, it serves as a starting point for further research. Future studies can build on these findings by incorporating additional metrics and exploring more intricate relationships. For instance, metrics derived from plantar pressure can be particularly valuable, as they closely relate to key kinematics and dynamics of the lower limb [36, 37]. Other useful metrics could be extracted from inverse dynamics [38] and potentially enhanced through machine learning techniques [39]. The present study provides valuable initial insights and establishes a basis for more comprehensive investigations into the interplay between personality traits and biomechanical performance of the lower limb.

The relationship between personality traits and movement patterns, as revealed by this research, hints at the potential for tailored physical interventions to enhance adherence and optimize responses in training programs. The connection between the information processing preference (the intuition-sensing axis) and RP could have potential implications for sports performance. As practical recommendations, we advocate for athletes to incorporate mental imagery alongside their physical practice, given its demonstrated ability to enhance performance [40]. In this context, we propose the mental engagement with abstract concepts as a strategy to amplify the intuition personality trait. This, in turn, is anticipated to enhance the plyometric capacity of the lower limb, ultimately increasing RP through the top-down process of embodied cognition.

This study presents several limitations. MBTI is seen as a controversial approach [41]. Indeed, several authors questioned its validity [42] and demonstrated psychometric limitations [43, 44]. Nonetheless, MBTI remains the most widely used personality assessment [24, 45]. The MBTI exhibits a strong correlation with the Neuroticism, Extraversion, Openness (NEO) Personality Inventory, an extensively employed personality assessment tool that scrutinizes the Big Five personality traits [46, 47]. In this study, MBTI was preferred over the Big Five. The reason is mainly due to the nuanced nature of the MBTI [13]. Briefly, personality traits are assigned among two distinct categories for each axis by MBTI while the Big Five indicates the absence or presence of a given personality trait. We believe that the MBTI assigns personality traits to each axis without implying superiority of one category compared to the other while

we believe that the Big Five tends to involve value judgments. Recent applications of the MBTI have delved into characterizing personality profiles for talent identification among elementary and junior high school students [48] and students within the broader population engaged in recreation and leisure [49], as well as among middle-aged male runners [50]. Additionally, the present findings are constrained by the exclusive use of the five-repetition rebound jump test to determine the RP of the lower limb, which could be seen as a subjective choice by the authors. Alternative tests, such as the countermovement jump and repetitive rebound in time to a metronome, could have been utilized to assess RP. However, research indicates that the countermovement jump test, due to extended $t_c$ and lack of pre-activation, reflects slow rather than fast stretch-shortening cycle function [51]. The repetitive rebound test to a metronome results in sub-maximal hopping. In comparison, the five-repetition rebound jump test yields a lower rebounding frequency, increased jump heights, and longer $t_c$ (attributable to greater angular displacements at the hips, knees, and ankles), along with reduced stiffness [52], hence higher RP. Furthermore, various confounding factors such as facial expression, emotion, and mood were not measured in this study. For example, perceived effort while running was shown to be higher when frowning compared to smiling and oxygen consumption lower when smiling compared to frowning [53]. Research also showed that anger could promote running symmetry while sadness could increase running asymmetry [54]. However, no scientific evidence showing that these factors influence the biomechanical variables measured herein were found. Additionally, the goal of the present study was not to account for sex differences (to maintain simplicity and ensure clarity), meaning that the recruitment process was not selective in terms of sex. Separating the genders would have also compromised statistical power. Nevertheless, considering the demonstrated but subtle differences in personality types between males and females [55], further research should prioritize examining the influence of sex on the relationship between the biomechanical performance of the lower limb and personality traits. Finally, because of the specific design of this study, conclusions cannot be predictive or causal. Nevertheless, the present study provides valuable initial insights into the relationship between the RP of the lower limb and personality traits.

## Conclusions

To conclude, the intricate connections between personality traits, as classified by the MBTI, and the biomechanical performance of the lower limb, assessed by the five-repetition rebound jump test, were unraveled by the present study. The investigation revealed a significantly larger RP for intuition than sensing individuals, indicating that intuition individuals displayed a greater reliance on the fast stretch-shortening cycle, showcasing superior use of their lower limb structures as springs. However, the study found no significant associations between biomechanical characteristics and the remaining three MBTI axes (extraversion-introversion, thinking-feeling, and judging-perceiving), challenging the initial hypothesis.

## Acknowledgments

We are grateful to the many volunteers of the present experiment. We thank Prof. Laurent Mourot (University of Franche-Comté) and Dr. Jean-Denis Rouillon for having initiated this study. The personality traits of participants was evaluated by Stephanie Giordano Assante (MBTI certified practitioner), which we thank for her extensive work.

## Author Contributions

**Conceptualization:** Cyrille Gindre, Thibault Lussiana.

**Data curation:** Aurélien Patoz, Bastiaan Breine.

**Formal analysis:** Aurélien Patoz, Bastiaan Breine.

**Investigation:** Cyrille Gindre, Thibault Lussiana.

**Methodology:** Cyrille Gindre, Thibault Lussiana.

**Supervision:** Cyrille Gindre, Thibault Lussiana.

**Writing – original draft:** Aurélien Patoz, Bastiaan Breine.

**Writing – review & editing:** Cyrille Gindre, Aurélien Patoz, Bastiaan Breine, Thibault Lussiana.

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
