## [Decision Letter · Decision Letter 0]

1 Jul 2024

PONE-D-24-17655Greater rebound power of the lower limb for intuition than sensing individualsPLOS ONE

Dear Dr. Patoz,

Thank you for submitting your manuscript to PLOS ONE. After careful consideration, we feel that it has merit but does not fully meet PLOS ONE’s publication criteria as it currently stands. Therefore, we invite you to submit a revised version of the manuscript that addresses the points raised during the review process.

We look forward to receiving your revised manuscript.

Kind regards,

Yaodong Gu

Academic Editor

PLOS ONE

2. Please upload a copy of S1 File. Personal protection committee EST I to which you refer in your text on page 14. Please amend the file type to 'Supporting Information'. If the Supplementary file is no longer to be included as part of the submission please remove all reference to it within the text.

Additional Editor Comments:

The methods part shall be more detailing described, it must prove to be reliable.

Reviewers' comments:

Reviewer's Responses to Questions

**Comments to the Author**

1. Is the manuscript technically sound, and do the data support the conclusions?

Reviewer #1: Yes

Reviewer #2: Partly

2. Has the statistical analysis been performed appropriately and rigorously? 

Reviewer #1: Yes

Reviewer #2: Yes

3. Have the authors made all data underlying the findings in their manuscript fully available?

Reviewer #1: Yes

Reviewer #2: Yes

4. Is the manuscript presented in an intelligible fashion and written in standard English?

Reviewer #1: Yes

Reviewer #2: Yes

5. Review Comments to the Author

Reviewer #1: Review comment

This manuscript entitled “Greater rebound power of the lower limb for intuition than sensing individuals” primarily aimed to examine the relation between personality traits

assessed by the MBTI test and the biomechanical differences in the RP of the lower limb. The results of this study provide guidance for human movement science and sports science. While it is a very interesting topic. But I think this manuscript has some flaws to fill in before it can be published in a journal. There are several questions should be addressed, which list below. I give a minor revision for this manuscript.

Specific comments

1. In the Abstract part, In the opinion of reviewer, the author provided too much background descriptions in this part, which may be too long-winded. I suggest that the authors provide more detailed descriptions of the methods, results, and conclusions of this study in this part.

2. In the introduction part, In the reviewer's opinion, lower limb biomechanics should include plantar pressure (which is closely related to key kinematics and dynamics of lower limbs). Therefore, the following studies are recommended for reference:


https://doi.org/10.1177/1754337120960962


https://doi.org/10.37190/ABB-01627-2020-02

3. “This question gains particular significance when considering tasks such as vertical jumps.” Please add a reference to support this sentence.

4. In the Materials and Methods part, “Eighty individuals, 13 females (age: 30 ± 12 years, height: 167 ± 7 cm, body mass: 61 ± 9 kg, and training hour per week: 9 ± 8 h/week) and 67 males (age: 29 ± 11 years, height: 178 ± 6 cm, body mass: 72 ± 9 kg, and training hours per week: 6 ± 4 h/week), voluntarily participated in this study.” How did the authors determine the sample size of females and males?

5. In the Discussion part, “This investigation unveiled that a penchant for abstract concepts correlates with a larger RP, indicating a more dynamic pattern.” Can you be more specific about what you mean for dynamic pattern.

6. In the Conclusion part, In the opinion of the reviewer, the description in the conclusion part was too verbose, and the reviewer suggests that the authors should abbreviate the section and focus on the main findings of this study.

Reviewer #2: The study is interesting, however, the connect for how personality traits related to biomechanical functions, specifically vertical jumping is not strongly illustrated.

In your study, participants were instructed to perform six consecutive hops, and repetitive performance (RP) was assessed based on these hops. Could you elaborate on why you chose to measure six consecutive hops rather than selecting the trial with the highest jump height from a session with multiple trials? Specifically, how does this approach contribute to the reliability and validity of your findings in the context of assessing RP?

Given that the MBTI is a controversial tool with documented psychometric limitations and questions regarding its validity, can you elaborate on why you chose to use the MBTI for assessing personality traits in your study? How do you address the concerns about its subjectivity and limited indices in the context of evaluating the relationship between lower limb repetitive performance (RP) and personality traits?

The results presented seem to indicate simple relationships between MBTI axes and biomechanical performance measures such as flight time () and repetitive performance (RP). Given the potential for more complex interactions between biomechanical performance and personality traits, could you elaborate on why the study focused on these specific measures? Additionally, how do you address the possibility that the simplicity of these results may not fully capture the nuanced relationships between personality and performance?

6. PLOS authors have the option to publish the peer review history of their article (what does this mean?). If published, this will include your full peer review and any attached files.

Reviewer #1: No

Reviewer #2: **Yes: **Liangliang Xiang

---

## [Author Response · Author response to Decision Letter 0]

11 Jul 2024

Subject: Response to the editor and reviewer

Manuscript title: Personality in Motion: How intuition and sensing personality traits relate to lower limb rebound performance

We would like to thank the reviewers for the useful comments. The feedback provided was helpful in improving the quality of our manuscript during the revision process. We have adapted our manuscript as suggested based on the reviewers’ comments. We have provided an answer to each comment. 

In addition, to facilitate the review process, we have indicated all modifications in the manuscript in RED font color, except for where we removed text. We hope that the reviewers find our responses meet their expectations.

Editor

2. Please upload a copy of S1 File. Personal protection committee EST I to which you refer in your text on page 14. Please amend the file type to 'Supporting Information'. 

Response: A copy of S1 File “Personal protection committee EST I” has been uploaded.

3. The methods part shall be more detailing described, it must prove to be reliable.

Response: Following the editor's suggestion, the methods section has been expanded to provide a more detailed and reliable description.

Lines 138-144

Then, RP, i.e., the plyometric characteristics of the lower limbs [21, 22], was assessed by asking participants to perform the five-repetition rebound jump test. All jumps were performed with hands placed on the hips and participants were wearing their habitual training shoes. Participants were required to jump vertically as high and as fast as possible while minimizing ground contact time (tc) and maximizing flight time (tf). Participants were also instructed to minimize knee actions (i.e., flexion and extension) during the test.

Lines 151-160

The Optojump® system, connected to a personal computer, consists of two parallel bars placed 1 m apart. One bar contains light-emitting diodes positioned 3 mm above the ground, acting as the transmitter unit, while the other bar serves as the receiver unit. Every millisecond (sampling frequency of 1000 Hz), the system checks the status of the light transmission. If the light is blocked by an individual’s feet during jumping, a designated variable is set to 1. Conversely, if the light reaches the receiver, the variable is set to 0. Therefore, tc (ms) was given by the time during which the light was blocked (variable is 1), and tf (ms) by the time during which the light was transmitted (variable is 0). Then, RP for each hop was calculated following the method described in Bosco, Luhtanen (22) using the following equation (Eq. 1).

Lines 162-167

Three repetitions of the test were performed with a 30 s rest. The best repetition, based on the highest RP was used for statistical analysis, to ensure the reliability and validity of the present findings in assessing RP. This approach allowed selecting the most representative performance data, minimizing the influence of variability that may occur in individual trials and provided a more accurate reflection of participants' lower limb biomechanics during vertical jumping.

Reviewer 1

This manuscript entitled “Greater rebound power of the lower limb for intuition than sensing individuals” primarily aimed to examine the relation between personality traits

assessed by the MBTI test and the biomechanical differences in the RP of the lower limb. The results of this study provide guidance for human movement science and sports science. While it is a very interesting topic. But I think this manuscript has some flaws to fill in before it can be published in a journal. There are several questions should be addressed, which list below. I give a minor revision for this manuscript.

Response: We thank the reviewer for the general positive feedback. The answer to each specific comment can be found below.

Specific comments

1. In the Abstract part, In the opinion of reviewer, the author provided too much background descriptions in this part, which may be too long-winded. I suggest that the authors provide more detailed descriptions of the methods, results, and conclusions of this study in this part.

Response: The reviewer's feedback on the abstract has been duly noted. Accordingly, the background descriptions have been streamlined, and greater emphasis has been placed on providing detailed descriptions of the methods, results, and conclusions of the study in the abstract section.

Lines 41-62

Embodied cognition asserts a symbiotic relationship between cognitive processes and the physical body, raising an intriguing question: could personality traits be intertwined with the biomechanical performance of the lower limb? This study aimed to explore this connection by examining how personality traits, assessed using the Myers-Briggs Type Indicator (MBTI), relate to lower limb rebound power (RP) measured through the five-repetition rebound jump test. Eighty participants completed two sessions: a biomechanical analysis of hopping using an Optojump® system to measure contact time, flight time, and RP, and a personality traits assessment categorizing traits across four MBTI axes: extraversion-introversion (favorite world); sensing-intuition (information processing preference); thinking-feeling (decision making); and judging-perceiving (structure). Participant characteristics did not significantly differ across MBTI axes (p�0.07), minimizing potential confounding factors. Notably, individuals classified as intuitive showed significantly longer flight times (p=0.02) and larger RP (p=0.007) compared to sensing individuals, suggesting a greater reliance on the fast stretch-shortening cycle and showcasing superior use of their lower limb structures as springs. This suggests potential implications for sports performance, with intuition individuals possibly excelling in plyometric sports. However, no significant associations were found between biomechanical performance and the other three MBTI axes (p≥0.12), challenging the initial hypothesis. This research provides initial insights into the nuanced relationship between personality traits and movement patterns, indicating the potential for tailored physical interventions to enhance adherence and optimize responses in training programs.

2. In the introduction part, In the reviewer's opinion, lower limb biomechanics should include plantar pressure (which is closely related to key kinematics and dynamics of lower limbs). Therefore, the following studies are recommended for reference:

https://doi.org/10.1177/1754337120960962

https://doi.org/10.37190/ABB-01627-2020-02

Response: The suggested studies on plantar pressure are more appropriately discussed in the section focused on future directions, in the discussion section of the manuscript. Here, additional metrics, such as those derived from plantar pressure, are highlighted for their relevance to understanding the key kinematics and dynamics of the lower limb. This discussion also encourages the exploration of other metrics from inverse dynamics, underscoring the study's contribution to advancing research on the interplay between personality traits and biomechanical performance of the lower limb.

Lines 306-308

For instance, metrics derived from plantar pressure can be particularly valuable, as they closely relate to key kinematics and dynamics of the lower limb [36, 37]. 

3. “This question gains particular significance when considering tasks such as vertical jumps.” Please add a reference to support this sentence.

Response: The requested reference has been added to support the statement regarding the significance of tasks such as vertical jumps.

Lines 101-103

Vertical jumps, widely used by strength and conditioning professionals, coaches, and healthcare practitioners, serve as performance tests for evaluating lower limb muscular strength [16], highlighting the significance of this question.

4. In the Materials and Methods part, “Eighty individuals, 13 females (age: 30 ± 12 years, height: 167 ± 7 cm, body mass: 61 ± 9 kg, and training hour per week: 9 ± 8 h/week) and 67 males (age: 29 ± 11 years, height: 178 ± 6 cm, body mass: 72 ± 9 kg, and training hours per week: 6 ± 4 h/week), voluntarily participated in this study.” How did the authors determine the sample size of females and males?

Response: The sample sizes of females and males in the study were determined by the voluntary participation of individuals who met the inclusion criteria, without selective recruitment based on sex. The decision not to separate genders was made to maintain simplicity and ensure clarity in the study design, as well as to avoid compromising statistical power. This decision has been emphasized in the limitations of the present study. 

Lines 356-362

Additionally, the goal of the present study was not to account for sex differences (to maintain simplicity and ensure clarity), meaning that the recruitment process was not selective in terms of sex. Separating the genders would have also compromised statistical power. Nevertheless, considering the demonstrated but subtle differences in personality types between males and females [55], further research should prioritize examining the influence of sex on the relationship between the biomechanical performance of the lower limb and personality traits.

5. In the Discussion part, “This investigation unveiled that a penchant for abstract concepts correlates with a larger RP, indicating a more dynamic pattern.” Can you be more specific about what you mean for dynamic pattern.

Response: We thank the reviewer for this comment. By "dynamic pattern," it is specifically meant that a larger rebound power (RP) can be achieved by reducing contact time (tc) and/or increasing flight time (tf), as outlined in Equation (1). This statement has been clarified in the discussion section of the updated manuscript.

Lines 273-274

According to Eq. (1), a larger RP can be obtained by decreasing t_c and/or increasing t_f, indicating a more dynamic rebound pattern.

6. In the Conclusion part, In the opinion of the reviewer, the description in the conclusion part was too verbose, and the reviewer suggests that the authors should abbreviate the section and focus on the main findings of this study.

Response: The conclusion section has been revised to abbreviate the description and emphasize the main findings of the study, as suggested by the reviewer.

Lines 368-376

To conclude, the intricate connections between personality traits, as classified by the MBTI, and the biomechanical performance of the lower limb, assessed by the five-repetition rebound jump test, were unraveled by the present study. The investigation revealed a significantly larger RP for intuition than sensing individuals, indicating that intuition individuals displayed a greater reliance on the fast stretch-shortening cycle, showcasing superior use of their lower limb structures as springs. However, the study found no significant associations between biomechanical characteristics and the remaining three MBTI axes (extraversion-introversion, thinking-feeling, and judging-perceiving), challenging the initial hypothesis. 

Reviewer 2

The study is interesting, however, the connect for how personality traits related to biomechanical functions, specifically vertical jumping is not strongly illustrated.

In your study, participants were instructed to perform six consecutive hops, and repetitive performance (RP) was assessed based on these hops. 

Could you elaborate on why you chose to measure six consecutive hops rather than selecting the trial with the highest jump height from a session with multiple trials? 

Specifically, how does this approach contribute to the reliability and validity of your findings in the context of assessing RP?

Response: As correctly pointed out by the reviewer, the explanation about the number of trials performed and about the selection criteria to obtain the rebound power (RP) metrics was missing in the manuscript. We have addressed this concern in the updated version of the manuscript. Specifically, participants performed three repetitions of the five-repetition rebound jump test with a 30-second rest interval between each repetition. The decision to use the best repetition, based on the highest RP, for statistical analysis was made to ensure the reliability and validity of our findings in assessing RP. This approach allows us to select the most representative performance data, minimizing the influence of variability that may occur in individual trials and providing a more accurate reflection of participants' lower limb biomechanics during vertical jumping.

Lines 162-167

Three repetitions of the test were performed with a 30 s rest. The best repetition, based on the highest RP was used for statistical analysis, to ensure the reliability and validity of the present findings in assessing RP. This approach allowed selecting the most representative performance data, minimizing the influence of variability that may occur in individual trials and provided a more accurate reflection of participants' lower limb biomechanics during vertical jumping.

Given that the MBTI is a controversial tool with documented psychometric limitations and questions regarding its validity, can you elaborate on why you chose to use the MBTI for assessing personality traits in your study? 

Response: The decision to use the MBTI for assessing personality traits in our study was based on several factors. The most important one is that the MBTI was chosen over the Big Five primarily due to its nuanced approach in categorizing personality traits into distinct types for each axis, as opposed to simply indicating the presence or absence of a trait. These factors have been described in the limitation section of the manuscript.

Lines 328-339

The MBTI exhibits a strong correlation with the Neuroticism, Extraversion, Openness (NEO) Personality Inventory, an extensively employed personality assessment tool that scrutinizes the Big Five personality traits [46, 47]. In this study, MBTI was preferred over the Big Five. The reason is mainly due to the nuanced nature of the MBTI [13]. Briefly, personality traits are assigned among two distinct categories for each axis by MBTI while the Big Five indicates the absence or presence of a given personality trait. We believe that the MBTI assigns personality traits to each axis without implying superiority of one category compared to the other while we believe that the Big Five tends to involve value judgments. Recent applications of the MBTI have delved into characterizing personality profiles for talent identification among elementary and junior high school students [48] and students within the broader population engaged in recreation and leisure [49], as well as among middle-aged male runners [50].

How do you address the concerns about its subjectivity and limited indices in the context of evaluating the relationship between lower limb repetitive performance (RP) and personality traits?

Response: In addressing concerns about subjectivity and limited indices in evaluating the relationship between lower limb rebound power (RP) and personality traits, it is important to acknowledge the methodological choices made in our study. Specifically, the exclusive use of the five-repetition rebound jump test to assess RP may be perceived as subjective. However, this test was selected based on its ability to provide insights into fast stretch-shortening cycle function, which is crucial for evaluating lower limb biomechanical performance. Alternative tests, such as the countermovement jump or repetitive rebound to a metronome, could have been considered. Yet, research suggests that the countermovement jump may not effectively capture fast stretch-shortening cycle function due to extended contact times and lack of pre-activation. Similarly, repetitive rebound to a metronome often results in sub-maximal hopping, which may not fully reflect maximal muscular performance. The revised manuscript now includes several sentences addressing this concern in its limitation section.

Lines 340-349

Additionally, the present findings are constrained by the exclusive use of the five-repetition rebound jump test to determine the RP of the lower limb, which could be seen as a 

---

## [Decision Letter · Decision Letter 1]

26 Aug 2024

Personality in Motion: How intuition and sensing personality traits relate to lower limb rebound performance

PONE-D-24-17655R1

Dear Dr. Patoz,

We’re pleased to inform you that your manuscript has been judged scientifically suitable for publication and will be formally accepted for publication once it meets all outstanding technical requirements.

Kind regards,

Yaodong Gu

Academic Editor

PLOS ONE

Additional Editor Comments (optional):

Well done!

Reviewers' comments:

Reviewer's Responses to Questions

**Comments to the Author**

1. If the authors have adequately addressed your comments raised in a previous round of review and you feel that this manuscript is now acceptable for publication, you may indicate that here to bypass the “Comments to the Author” section, enter your conflict of interest statement in the “Confidential to Editor” section, and submit your "Accept" recommendation.

Reviewer #1: (No Response)

Reviewer #2: All comments have been addressed

2. Is the manuscript technically sound, and do the data support the conclusions?

Reviewer #1: (No Response)

Reviewer #2: Partly

3. Has the statistical analysis been performed appropriately and rigorously? 

Reviewer #1: (No Response)

Reviewer #2: Yes

4. Have the authors made all data underlying the findings in their manuscript fully available?

Reviewer #1: (No Response)

Reviewer #2: Yes

5. Is the manuscript presented in an intelligible fashion and written in standard English?

Reviewer #1: (No Response)

Reviewer #2: Yes

6. Review Comments to the Author

Reviewer #1: Thank you to the authors for their additional efforts. The reviewer believes that, after revisions, this manuscript has reached the standard for publication.

Reviewer #2: (No Response)

7. PLOS authors have the option to publish the peer review history of their article (what does this mean?). If published, this will include your full peer review and any attached files.

Reviewer #1: **Yes: **Zixiang Gao

Reviewer #2: No

---

## [Editor Report · Acceptance letter]

23 Sep 2024

PONE-D-24-17655R1 

PLOS ONE

Dear Dr. Patoz, 

I'm pleased to inform you that your manuscript has been deemed suitable for publication in PLOS ONE. Congratulations! Your manuscript is now being handed over to our production team.

Kind regards, 

on behalf of

Professor Yaodong Gu 

Academic Editor

PLOS ONE